# Human T-Cell Leukemia Virus Type 1-Related Diseases May Constitute a Threat to the Elimination of Human Immunodeficiency Virus, by 2030, in Gabon, Central Africa

**DOI:** 10.3390/v14122808

**Published:** 2022-12-16

**Authors:** Eldridge Fedricksen Oloumbou, Jéordy Dimitri Engone-Ondo, Issakou Mamimandjiami Idam, Pamela Moussavou-Boudzanga, Ivan Mfouo-Tynga, Augustin Mouinga-Ondeme

**Affiliations:** Unité des Infection Rétrovirales et Pathologies Associées, Centre International de Recherches Médicales de Franceville (CIRMF), Franceville BP 769, Gabon

**Keywords:** HIV-1, HTLV-1, coinfection, genetic diversity, HTLV-1-associated pathologies, HIV elimination, Gabon

## Abstract

The Joint United Nations Program on HIV/AIDS (UNAIDS) has adopted the Sustainable Development Goals (SDGs) to end the HIV/AIDS epidemic by 2030. Several factors related to the non-suppression of HIV, including interruptions of antiretroviral therapy (ART) and opportunistic infections could affect and delay this projected epidemic goal. Human T-Cell leukemia virus type 1 (HTLV-1) appears to be consistently associated with a high risk of opportunistic infections, an early onset of HTLV-1 and its associated pathologies, as well as a fast progression to the AIDS phase in co-infected individuals, when compared to HIV-1 or HTLV-1 mono-infected individuals. In Gabon, the prevalence of these two retroviruses is very high and little is known about HTLV-1 and the associated pathologies, leaving most of them underdiagnosed. Hence, HTLV-1/HIV-1 co-infections could simultaneously imply a non-diagnosis of HIV-1 positive individuals having developed pathologies associated with HTLV-1, but also a high mortality rate among the co-infected individuals. All of these constitute potential obstacles to pursue targeted objectives. A systematic review was conducted to assess the negative impacts of HTLV-1/HIV-1 co-infections and related factors on the elimination of HIV/AIDS by 2030 in Gabon.

## 1. Introduction

The human immunodeficiency virus/acquired immune deficiency syndrome (HIV/AIDS) pandemic reduces the availability of human capital and negatively affects the economic growth worldwide [1]. HIV/AIDS has impacted the labor supply by increasing mortality, morbidity, and expenditure on one hand, and reduced revenues, on the other hand [2,3]. In Sub-Saharan Africa, HIV/AIDS has limited the workforce and the agricultural productivity, increased poverty, and adversely changed the societal tissue and familial structure, leading to mono-parenting or kinship care. The HIV/AIDS-related stigmatization and discrimination constitute additional challenges for people living with these conditions [4]. The sustainable development goals aim to reconcile the economic growth, environmental balance, and social progress by ensuring that all people have equal opportunities and limited damage to the planet. UNAIDS adopted the Sustainable Development Goals (SDGs) to end the HIV/AIDS epidemic by 2030 (Resolution A/RES/70/1: Transforming our world: the 2030 Agenda for Sustainable Development; New York; September 2015). However, several factors are associated with the HIV non-suppression, including antiretroviral therapy (ART) interruptions and severe opportunistic infections that could interfere with treatments and delay the ending of this epidemic, as intended [5]. In the absence of ART, the development of opportunistic infections and the rapid progression to the AIDS phase, as seen in the case of HIV-1 and human T-cell leukemia virus type-1 (HTLV-1) co-infections could be a major setback for the proper management of the condition [6]. Both of these retroviruses share transmission routes and have a common in vivo tropism, since the CD4+ T-cells are their major targets of infection [7]. HTLV-1 is the etiological agent of several pathologies, such as a very severe T-cell lymphoproliferation, named adult T-cell leukemia/lymphoma (ATLL), and a progressive disabling neuro-myelopathy, known as the tropical spastic paraparesis/HTLV-1 associated myelopathy (TSP/HAM) [8,9,10]. Several studies have shown that both viruses (HTLV-1 and HIV-1) are very prevalent in Gabon, revealing high prevalence rates in all provinces of the country [2,11]. Unfortunately, no research or adequate patient care program related to the HTLV-1-mediated diseases has been conducted, and HIV is circulating with an important genetic diversity. All programs conducted in the country did not allow to achieve the UNAIDS’ 90-90–90 goals by 2020. Similarly, the factors involved in the non-achievement of these goals could affect and prevent the elimination of HIV by 2030. Furthermore, the HIV-1/HTLV-1 co-infections have not yet been characterized nor have the HTLV-1-related diseases been described. We conducted a systematic review to describe the available findings on HIV-1 and HTLV-1 and evaluate the negative impacts of the HTLV-1/HIV-1 coinfections on the HIV elimination by 2030, in Gabon. The HTLV-2 infection is uncommonly related to a HTLV-1 infection in Gabon [11,12]. Furthermore, a HTLV-2 infection is known as a rarely occurring pathogenic condition, and a HTLV-2/HIV-1 co-infection has often been reported to be associated to a low viral burden of HIV-1, thus a very low progression to the AIDS phase [13,14]. Henceforth, in this review, we have not discussed the aspects related to a HTLV-2 infection and its co-infection with HIV-1.

## 2. Epidemiological Situation of the HIV-1 Infection in Gabon

### 2.1. HIV-1 Prevalence

An estimated 38.4 million people are affected by HIV-1, with approximately 1.5 million new infections diagnosed annually, accounting for 650,000 deaths, according to the UNAIDS report. The epidemiological situation of HIV-1 worldwide, by region, showed that the region of sub-Saharan Africa was the most affected with approximately 25.6 million people living with HIV (PLHIV) for 860,000 new cases and 420,000 AIDS-related deaths annually [1]. In Gabon, a central African country, the last Demographic and Health Survey (DHS) reported 47,000 infected people, and the prevalence was estimated at 4.1% [2]. Specifically, 5.8% of women and 2.2% of men were infected with HIV-1 in 2012. Women living in urban areas (5.9%) seemed to be more infected than those in rural areas (5.3%). In contrast, men in rural areas (2.7%) were more infected than those in urban areas (2.1%) [2]. However, regarding the 2020 UNAIDS report on the progress of the fight against HIV/AIDS, the volume of HIV-positive screening in Gabon was estimated at 8.3% out of 135,093 tests carried out nationwide [3]. These findings suggested an increase of the HIV prevalence in Gabon, since 2012. This implies that if another study is carried out on the prevalence of the HIV infection in Gabon, the results would be higher than that of the 4.1% published in the last survey in 2012, in Gabon. In addition, the HIV seroprevalence varies by province (Figure 1), the Ogooué-Ivindo province showed the lowest seroprevalence, with 3.3% among women, versus 1.3% for men. Conversely, the Woleu Ntem province had the highest seroprevalence of 9.7% for women and 4.6% for men [2]. According to UNAIDS, the incidence of HIV-1 has significantly dropped from 1.8% in 2010 to 0.67% in 2020. This has also led to a drop in the mortality rate with the total number of deaths due to AIDS out of a population of 100,000 people, decreasing from more than 100 deaths in 2010 to almost 50 deaths in 2020 [3]. The data on the key populations in Gabon are quite unknown due to the stigmatization and discrimination associated with HIV infections.

### 2.2. Genetic Diversity of HIV-1

The HIV-1 infection in Gabon is characterized by an important genetic diversity that includes the circulating groups, sub-types, and all recombinant forms of HIV. As group M is the most circulating in the country, the sub-type A, and the recombinant form CRF02_AG were found to be predominant. These HIV-1 variants were described by a general population, specifically in Libreville and Franceville, the capital city of Gabon and the province of Estuary, and the capital of the province of Haut-Ogooué (southeast), respectively [15]. These findings were confirmed and 49% of the sub-type A seroprevalence was estimated in the general population [15]. Then, two years after that, another study found that the CRF02_AG sub-type was predominant in the general population of Gabon, at 26% with other recombinant forms at 19% [16]. In 2008, Caron et al. highlighted a predominant circulation of the CRF02_AG sub-type at 57% followed by the A sub-type at 9% among the miners of COMILOG, a company located in Moanda in the Haut-Ogooué province [17]. In 2009, a strong circulation of CRF02_AG was reported in the general population of the cities of Makokou and Oyem, the capital cities of the provinces of Ogooué-Ivindo, and Woleu-Ntem, respectively [18]. In 2012, Caron et al. carried out a study with the National Program for the Fight against HIV in pregnant women, and the results still confirmed the high genetic diversity of HIV-1 in Gabon. A predominance of CRF02_AG at 46.7% followed by subtypes A at 19.6%, G at 10.3%, F at 4.7%, M at 1.9%, and D at 0.9%, were obtained [19]. Studies conducted in Franceville and Koulamoutou (capital of the province of Ogooué-Lolo) between 2013 and 2022 demonstrated the development of resistance to antiretroviral treatments, and the increasing genetic diversity with various recombinant forms circulating, including the CRF06_cpx, CRF09_cpx, CRF45_cpx, CRF11_cpx, CRF37_cpx, and CRF49_cpx [20,21,22]. Among the other HIV-1 groups (N, O, and P) circulating in Gabon, group O was identified for the first time in 1986, in Libreville, and two other cases were reported in 2013 [23,24]. So far, no study has been conducted to define the HIV-1 variants circulating in the province of Moyen-Ogooué. Figure 1 presents the prevalence and genetic diversity of the circulating variants, but an all-inclusive study must be carried out to confirm the percentages per province of HIV circulating forms in Gabon. These circulating forms are not exclusive to Gabon, as they were equally identified in neighboring countries, as presented in Table 1 [24,25,26,27,28,29,30]. In Gabon, since the study by Delaporte et al. in 1996 reported a low seroprevalence of HIV-2 (3.5% or 8/226 samples) [24], no other study has focused on the prevalence of HIV-2. Furthermore, knowing that Gabon is located in Central Africa where the M group of HIV-1 predominates, all the studies carried out in this country have exclusively focused on the HIV-1 infection [25,26].

### 2.3. Diagnosis of HIV-1 in Gabon and the Therapeutic Management

The HIV screening in Gabon is based on the performance of several serological laboratory tests. The tests are performed according to the recommendations of the national program for the fight against HIV/AIDS and sexually transmitted infections (STIs, PLIS), thus combining the detection of antibodies directed against HIV-1 and HIV-2, and the detection of antigen (Ag) p24. The recommended tests are the DETERMINE HIV1/2, Alere COMBO HIV, and SD Bioline HIV-1/2 3.0 [25]. These rapid diagnostic tests use the principle of immunochromatography with biological material, such as whole blood, plasma, and serum.

According to the manufacturers, all three tests have 100% sensitivity and 99.8% specificity levels. As per the manufacturer’s recommendations, these tests can detect all HIV-1 subtypes, which are predominant in Gabon. A recent study carried out in Libreville, evaluated the Alere COMBO HIV and DETERMINE HIV1/2 tests, and the results confirmed the stipulated sensitivity while the specificity for Alere COMBO HIV and DETERMINED HIV1/2 were 94.6% and 98.6%, respectively [28]. Based on the specificity values, these tests were excluded from the HIV screening algorithm in Gabon following the WHO recommendations, which stipulate that screening tests must have a specificity greater than or equal to 98% [29]. However, a study evaluating these screening tests must be extended nationwide to bring precision on their reliability, as well as other tests which are not yet included in the program. Furthermore, with the high genetic diversity present in Gabon, it would be important to show that the tests included in the national screening program make it possible to diagnose all of the circulating subtypes. For therapeutic care, the WHO recommendations are systematically applied in Gabon, and the newly HIV-diagnosed patients are immediately put on antiretroviral treatment, regardless of the CD4 levels [30]. Currently, the complete and sustained virological suppression relies on triple therapy, a combination of three active antiretroviral agents to ensure the full potency and avoid the selection of resistant viruses from the viral quasi-species [31]. Until late 2019, the first line of antiretroviral treatment in Gabon consisted of reverse transcriptase inhibitors, which are made up of two nucleoside (or nucleotide) reverse transcriptase inhibitors (NRTIs) and one non-nucleoside inhibitor (NNRTI). The referential treatment was tenofovir disoproxil fumarate (TDF) + lamivudine (3TC)/emtricitabine (FTC) + efavirenz (EFV)/nevirapine (NVP) [32]. Following reports of an increased pre-treatment resistance to the non-nucleoside reverse transcriptase inhibitors (NNRTIs) in low- and middle-income countries in 2019, the WHO recommended dolutegravir (DTG)-containing regimens as the preferred first- and second-line antiretroviral therapy (ART) for PLHIV [33]. In 2012, two major mutations were reported in PLHIV in the regions of Franceville and Koulamoutou, and indicated 17 and 37.6% for the NNRTI and the reverse transcriptase inhibitor mutation rates, respectively. A recent study conducted in 2021 reported a 21.9% reverse transcriptase inhibitor mutation rate in the same regions of Franceville and Koulamoutou [20,21]. The country treatment strategy of PLHIV was readjusted following the WHO recommendations and local realities, and thus the first line of treatment combined two NRTIs (TDF + 3TC/ABC + 3TC) + DTG [27]. The NRTIs are derivatives of natural nucleosides, and those that are included in the treatment of PLHIV in Gabon are tenofovir (TDF), abacavir (ABC), lamivudine (3TC), and emtricitabine (FTC). They can be considered as prodrugs because they undergo intracellular triphosphorylation leading to the active derivative of TI. They essentially act in competition with natural nucleosides by blocking the synthesis of the proviral DNA by reverse transcriptase [34,35,36]. DTG belongs to one of the newer classes of antiretroviral (integrase inhibitors) drugs approved by the FDA for the treatment of the HIV infection. Integrase inhibitors irreversibly inhibit HIV integrase and prevent the integration of the viral DNA into the host DNA leading to the inhibition of the provirus formation and the viral spread [37].

### 2.4. Treatment-Related Challenges and Complications Associated with HIV-1 in Gabon

The Joint United Nations Program on HIV/AIDS (UNAIDS) has set goals of 95-95-95 targets for eliminating HIV/AIDS by 2030. The final 95 means that 95% of people on an ART with a suppressed viral load, is only possible if 95% of PLHIV are on ART. Indeed, treatment is the main mean by which these goals can be achieved [33]. However, complications may occur leading to prevention of the 95-95-95 targets and subsequently the non-elimination of HIV/AIDS by 2030. The main complications associated with ART are the poor compliance with treatment, resistance to ART, and co-infections with other retroviruses. Adherence to ART is a primary determinant of the treatment success. A high level of adherence to ART is necessary to achieve and sustain the viral suppression, thereby preventing morbidity and mortality [34]. Indeed, an at least 95% adherence has been shown to provide the continuous and complete viral suppression [35]. However, PLHIV does not always manage to achieve or maintain high levels of adherence to treatment and the patient- and program-related factors could be reasons of such a low adherence. The patient-related factors involve the incomplete compliance, difficulty paying for treatment, routine laboratory tests, toxicity and side effects of ART regimens, and other program-related factors, such as treatment availability [20]. When patients show a poor adherence to treatment, they are expected to develop treatment-resistant mutations [36]. The failure of treatment is sometimes associated with the appearance of resistant mutations to the molecules constituting ART and co-infection with other retroviruses, such as the human T-cell leukemia virus (HTLV) [37]. There is little evidence on the treatment failure at the national level to predict whether the triple therapy currently offered works or not. However, in rural and semi-rural areas, two studies have been carried out in the southeast province of Gabon. The semi-rural settings are located in the provincial capital city perimeters, where the study was carried out with a large and active filing of patients. While the rural area is located further from the study site and with the least number of treated patients. In 2012, the prevalence rate was 41.3% and closely similar to that of 2021, which was 42%. The objectives of these two studies were to define the efficacy of the first-line treatment administered in Gabon, by evaluating the virological success rate, and consequently, the rate of treatment resistance mutations. In both studies, the high rates of treatment failure were reported [20,21].

## 3. Human T-Cell Leukemia Virus Type 1 (HTLV-1) in Gabon

### 3.1. General Epidemiological Data

Early epidemiological studies on the HTLV-1 infection in Gabon were conducted by Delaporte et al. in 1986. The initial study group consisted of 2558 participants made of 1874 adults and 684 children from cities of five provinces including the northeast province of Ogooué-Ivindo, the southern province of Ngounié, the southeastern province of Haut-Ogooué, and the major cities of Libreville and Port-Gentil of the coastal and western provinces of Estuary and Ogooué-Maritime, respectively [38]. An indirect ELISA test (to detect IgG against the HTLV-1 proteins) was used as an orientation test and a western blot as a confirmation test for the infection. The western blot test was declared positive when there was a p19 and/or p24 reactivity [38]. The reported seroprevalence ranged from 5% to 10.5% HTLV-1 in adults, versus 2% to 2.4% in children. Thus, a high seroprevalence was recorded and ranged between 9.4% to 10.5% in Ngounié and Haut-Ogooué provinces. This pioneering study on the HTLV-1 infection in Gabon induced a particular interest among researchers for this retrovirus that was mainly endemic to the southeastern provinces of Gabon. To further characterize the HTLV-1 epidemiology, several studies were carried out in the province of Haut-Ogooué. In 1991, Schrijvers et al. simultaneously assessed both the retroviral HIV and HTLV-1 prevalence in women of childbearing age, according to their fertility status. A total of 654 representative samples were collected from eight departments and 27 villages around the Franceville area. The estimates of the HIV and HTLV-1 seroprevalence were determined at 1.4% and 6.8%, respectively, with two cases of HIV-1/HTLV-1 co-infections confirmed after a western blot [39]. There was no peculiarity between the HTLV-1 infection and infertility, but the study still found an increasing HTLV-1 seroprevalence with age. The different HTLV-1 seroprevalences found in both studies conducted in Haut-Ogooué province by Delaporte et al. and Schrijvers et al. were 9.4% and 6.3%, respectively [38,39]. Furthermore, the main difference between the two studies was that one comprised only women while the other simultaneously encompassed individuals of both sexes. Despite the difference between the two seroprevalences, the studies have demonstrated a remarkable endemicity for the HTLV-1 infection in the province of Haut-Ogooué.

Additional studies were performed and confirmed the previous findings. In 1993, the first epidemiological data on the HTLV-1 infection were obtained after analyzing blood donors and pregnant women collected at the Regional Hospital of Franceville, Gabon. Six-hundred-and-thirty-three pregnant women and seven-hundred- and four donors were tested with ELISA and the positive samples were confirmed by a western blot (Dupont de Nemours). The HTLV-1 positivity was defined as the presence of gag (p19 with/without p24) and the envelope glycoprotein gp46. The western blot tests were performed with less serologically stringent criteria positivity, to better compare the characteristics with those of other studies previously conducted in the same region. Using less serologically stringent criteria lowered the risk of having negative results for the possible HTLV-2 positives and/or other specific STLV variants from Gabon. The reactivity to p19 and/or p24 with at least one pX gene protein encoding for the tax protein (p38, p42) or gp46 was detected and the HTLV infection was confirmed with a seroprevalence rate of 9.3% among pregnant women and 10% in blood donors [40]. That survey on the HTLV infection among pregnant women seemed to be quantitatively and qualitatively related to the work of Schrijvers and coworkers, between September 1986 and January 1987, in the same area but conducted with women of childbearing age (15–54 years). With just a little difference in their studied populations (pregnant vs. childbearing aged women), there was a 3% of gap for the HTLV-1 infection prevalence, which came out from both studies with a 6.3% outcome for a HTLV-1 seroprevalence in the work of Schrijvers and coworkers [39,40]. The difference could be explained by the non-strict application of the HTLV serological criteria to the confirmation test in the Berteau et al. study. The hypothesis was justified by the 5.5% HTLV-1 seroprevalence rate that was very close to the 6.3% reported by Schrijvers et al. and obtained following the strict serological application criteria to the western blot confirmation test in Berteau’s et al. study [39,40].

An investigation by Hersan and coworkers aimed to identify the demographic and social factors involved in the high HTLV-1 seroprevalence observed in Haut-Ogooué province, demonstrated in 1994 with a predominance of the HTLV-1 infection in female individuals together with heterogeneous seroprevalence, according to the ethnics groups, and with the highest prevalence obtained in the Kota-Obamba group, unlike the Teke group (11% vs. 3.4%) [41]. The difference in prevalence implied a possible involvement of social-cultural behaviors in the spread and maintenance of the viral persistence in these groups. A recent study carried out in 2018 on social and risk factors associated with the HTLV-1 infection in Gabon included 2060 high-risk individuals (261 pygmies vs. 1797 Bantu) from six provinces. A significant association between the HTLV-1 infection and the pygmies group was shown with a high risk in this group, unlike the Bantu group [12]. The reasons that could explain the high prevalence of this retrovirus within some ethnic communities, unlike others, are still hypothetical. It is estimated that strong vertical transmissions associated with the practice of certain initiatory rites, together with marriages between individuals of the same ethnic group, could explain the high prevalence recorded within these ethnic communities. Furthermore, based on the 4th generation ELISA and the western blot confirmatory tests, together with the PCR (on *env* and *LTR* region), an overall prevalence of the HTLV-1 infection rate of 8.7% among the rural population was reported and linked to the 8.5% previously reported by Hlela and coworkers in 2009, after compiling the epidemiological HTLV-1 infection data in Gabon, from 1980 to 2007 [42]. The same study reported that the HTLV-1 seroprevalence ranged from 6.6% to 8.5% and the endemic character of this retrovirus was also confirmed in Gabon [39,41,43]. However, these previous epidemiological data seemed to strongly coincide with the most recent report by Djuicy and coworkers. Prior to 2007, all epidemiological studies on the HTLV-1 infection prevalence in Gabon were just limited to the ELISA and western blot tests. The work carried out by Delaporte and coworkers aimed to determine the HTLV-1 prevalence and also to evaluate the PCR efficacy, compared to the commonly used serological (ELISA and western blot) tests. Using the PCR on the *gag*, *pol*, and *pX* (coding for the *tax* and *rex* proteins) regions, the HTLV-1 prevalence rate was estimated at 3.7%, compared to 8.1% for the seroprevalence rate [44]. This could represent one of the first real data on the HTLV-1 infection prevalence in Gabon, but could not reflect the national situation as the study was conducted on cluster samples in urban areas. However, it still confirmed the relatively high prevalence of the HTLV-1 infection in Gabon. In a similar study focusing on pregnant women, an estimated prevalence rate of 2.1% was obtained using a set of methodological tests (ELISA, WB, Env, and LTR qPCR) [45]. This relatively low prevalence could be linked to the fact that this study has not considered the differences between the urban and rural areas.

One of the latest epidemiological studies on the HTLV-1 infection in Gabon, was conducted on 4381 individuals from 220 villages spread over all nine provinces of the country. The heterogeneity of the HTLV-1 prevalence, according to the provincial areas was defined as follows: a relatively low prevalence (2–5%), a strong prevalence (5–10%), and a very strong prevalence or endemic prevalence (rate ≥ 10%). From the serological tests (ELISA and western blot), together with the molecular tests through the PCR, an overall HTLV-1 prevalence rate was estimated at 7.3%, which was very close to 8.7%, reported by Djuicy and coworkers in the same year [11,12]. That very close prevalence could be explained by the similarities of the studied populations. Indeed, both studies were carried out on people living in rural areas but only from six provinces and they were the first to report an 8.7% prevalence rate. Thus, both studies were carried out on the HTLV-1 infection predominance and mostly involved the rural populations of the south, southeast, and eastern sides of the country, corresponding to Ngounié, Haut-Ogooué, and Ogooué-Ivindo provinces. Both Ogooué-Ivindo and Ogooué-Lolo provinces seemed to be competing for the most endemic province for this retrovirus with prevalence rates above 11% [11,12]. They are followed by Haut-Ogooué and Ngounié provinces occupying the 3rd and 4th places with a prevalence of 10% and 8%, respectively. Figure 2 below presents an overview of the prevalence rates obtained per province and related to their endemicities to the HTLV-1 infection, according to Djuicy et al., together with the work published by Caron et al., in 2018. Surprisingly, Moyen-Ogooué and Estuaire provinces had an identical prevalence rate of 6.7%. The provinces of Estuaire, Ogooué-Maritime, and Moyen-Ogooué were not considered by Djuicy and coworkers and their prevalence rates on the map were solely from the study of Caron et al. Overall, the two studies allowed for the better estimation of a viral infection with higher prevalence rates in the southeastern region of the country ranging from 8% to over 10% [11,12]. Possible reasons for the repartition of HTLV-1 are not yet elucidated but the zoonotic transmission is associated with high-risk factors and could be strongly envisaged. In comparison to the coastal regions, the southeastern countryside has a wide and wild forest cover zone, coupled with rural lifestyles, such as hunting, butchering, and eating wild animals, that are commonly practiced [11,12,46].

### 3.2. Modes of the HTLV-1 Transmission in Gabon

Beyond the studies based on the HTLV-1 epidemiology, many other possible factors associated with the HTLV-1 infection have been investigated, and the vertical transmission of this retrovirus is of particular interest. A particular study demonstrated that there is no association between the HTLV-1 infection and the infertility status, while the seroprevalence rate of the HIV-1 infection was mostly higher in women with primary infertility, than others (9.3% vs. 0.7%) [49]. These previous studies seemed to strengthen the work performed by Vill’s et al., a case-control study for 12 months that was published in 1991 and conducted on the HTLV-1 infection on 135 pregnant women and their babies, and the main conclusion indicated that the HTLV-1 infection did not significantly affect the progression or outcomes of the pregnancy, up to one year after delivery [50]. No child born from a seropositive mother presented a seroconversion after one year. Five years later, the same team published (in 1996) another work that assessed, for the first time in Gabon, the mother to child transmission of HTLV-1, particularly in the southeast of the country (Franceville). During these 4 years of follow up, 34 babies born of 45 women who were seropositive to HTLV (43 for HTLV-1 and two for HTLV-2) were considered, and a 17.5% risk of overall seroconversion (six positives babies) was reported and a HTLV-1 incidence seroconversion evaluated to 4.8 cases per 100 persons by year. To better highlight the mother to child transmission, the authors have searched the HTLV-1 by PCR in blood samples from the mother, the amniotic fluid and the cord blood. None of the samples of cord blood and amniotic fluid were positive for HTLV-1. In addition, the serological follow up of the babies indicated a disappearance of the mother’s HTLV antibodies from 6 to 12 months and the first seroconversion from 18 months, which led to a probable involvement of the vertical transmission in the spread of HTLV-1, in Gabon [51]. This finding was consistent with other studies from the same team, which described a 2.8% rate (17/610) of the HTLV-1 seroprevalence in children between 6 months and 14 years of age [51,52]. They found a 7.1% (31/434) of the HTLV-1 seroprevalence rate in mothers in Franceville, on average, and demonstrated that the HTLV-1 transmission from mother to child happens in the same proportions as through blood transfusion. Indeed, in this study, the authors showed a 19.4% risk of seroconversion (six positives babies for 31 positives mothers) against 19.2% (11/57) for the HTLV-1 infection for children born of seronegative mothers but who were diagnosed with sickle cell and regularly transfused with total blood [52]. However, other tested children who were seropositive for HTLV-1 and born of seronegative mothers did not confirm the preceding observation. Nevertheless, it must be noted that the 6% HTLV-1 seroprevalence rate described in 1993, by Berteau et al., in blood donors in the same town, seemed to strengthen these results [53]. From the seroprevalence data, the HTLV-1 infection among children from 0 to 5-years ofage seemed to present some similar epidemiologic characteristics described in other local studies, and no variation associated with sex or age was identified [54]. A further high relative seroprevalence rate in rural and inland areas (Ngounie, Haut-Ogooué). compared to the coastal and urban areas (Estuaire, Ogooué-Maritime) was particularly noticed in adults [54,55]. Some results obtained through the usual serological (ELISA and western blot) tests and confirmed through the molecular analyses (gag, env region amplification from proviral DNA), permitted to show that breastfeeding was a HTLV-1 transmission mean to children [51].

In addition to maternal transmission, blood transfusion could represent one of the major routes for spreading HTLV-1 in Gabon. Indeed, as it was said above, in 1993, Berteau et al. reported a 6% HTLV-1 seroprevalence rate among 704 blood donors in Franceville in Haut-Ogooué province. In 2017, a similar study was carried out in Libreville at the National Center of Blood Transfusion (CNTS), by Ramasamy et al., where they found a 0.74% HTLV-1 prevalence rate from 3123 blood donors [56]. Those studies on the HTLV-1 prevalence among blood donors are the only ones that indirectly offer insight into the risk of the HTLV-1 transmission through blood transfusion in Gabon. Despite the various epidemiological studies highlighting the high prevalence of this retroviral infection in Gabon, no routine diagnosis of HTLV-1 is performed at the CNTS nor in any other adequate structure in the country [40,56]. Thus, they highlighted the probable nosocomial acquisitions of the HTLV-1 infection, reinforcing the studies of the teams of Djuicy and Caron that described the significant associations between the HTLV-1 infection and multiple hospitalizations, as well as surgical operations [11,12]. Additionally, a rarer transmission route, which is the deep bite by non-human primates (NHPs), is mainly reported during hunting. HTLV-1 would come from its simian counterpart, the simian T lymphotropic virus type 1 (STLV-1), which is a retrovirus associated with the T-type leukemias in NHPs and is present in many species of monkeys [49,53,55]. In Gabon, the inter-species transmission from NHPs to humans seemed to occur mainly through NHP bites and, mostly during hunting parties, according to the study conducted by Kazanji and coworkers on 78 Gabonese living in rural environments and who were already bitten by a monkey [46]. The study reported an 8.97% prevalence rate, meaning seven positives out of 78 tested samples. This idea of the HTLV-1 transmission through deep bites by NHPs was confirmed by another study published one year later about the discovery of two new HTLV-4 strains that were isolated from hunter-bitten gorillas [57]. Otherwise, a significant association between the HTLV-1 infection and consumption of NHP meat has been highlighted in Caron et al.’s study [11]. However, the consumption of NHP meat that is not properly cooked, the NHP skinning and cutting carried out before the preparation of those meats, exposing the blood of the infected NPH to the people handling the NHP materials, could probably be a means of transmission. As described in Gabon, some inter-species transmission of STLV-1 from NPHs to humans by deep bites, have also reported in other countries, such as in Cameroon in 2015, by Filippone et al., where the authors have shown 8.6% (23/269) of the HTLV-1 prevalence with bitten people (during hunting parties) [58]. However, in 2017, a study conducted by Mousson et al. both on NHP bushmeat and humans from two different African regions (the Taï region, in the Cote d’Ivoire region and Bandundu, in the Democratic Republic of the Congo), have shown a high HTLV-1 prevalence for the Democratic Republic of the Congo 1.3% (*n* = 302) against 0.7% (*n* = 574) for the HTLV-1 prevalence in the Taï region, in the Cote d’Ivoire. This trend seems to be inversed for the STLV-1 infection for these two regions with 23% (*n* = 39) and 39% (*n* = 111), respectively, in each region. Remember that here, the aim of Mousson’s study et al. was to evaluate whether frequent contacts between human and NHP bushmeat infected by STLV-1, in the region, could be related to the HTLV-1 prevalence in the same region. Following a multivariate analysis, the authors highlighted the higher frequency butchering activities of NHPs in the Democratic Republic of the Congo, relative to the Cote d’Ivoire region. Furthermore, they reported two HTLV-1 strands very closely related to STLV-1, isolated from same region in the Cote d’Ivoire [59]. Thus, this study reinforces the findings in some studies conducted in Gabon, which showed an existence of the probable inter-species transmission through contacts and the NHP bushmeat consumption.

### 3.3. Cases of the Associated Disease with the HTLV-1 Infection Described in Gabon

Gabon is one of the most endemic countries for the HTLV-1 infection in the world, with about 16,000 to 30,000 infected people according, to Gessain and Cassar’s published work, in 2012 [60,61]. About 3 to 7% of HTLV-1-infected people will develop an associated pathology during their lifetime [50,61,62], thus at least 480 to 2100 cases of the HTLV-1-induced diseases should have already been reported (since 1986), as the number of infected people is probably underestimated [60]. Up until today, there is no study which shows the incidence to the HTLV-1 associated diseases. Only some rare leukemia or lymphoma and spastic paraparesis cases have been reported. The first cases of diseases related to HTLV-1 in Gabon, are the TSP/HAM cases, identified during an epidemiological investigation conducted by Delaporte et al. in 1989, in two men living in a village in Haut-Ogooué province [51]. Both sera-epidemiologic and clinical means were considered to diagnose the two patients, of about 55 and 60 years of age. Several local clinicians could not retain the diagnosis of TSP/HAM, which is not commonly used and the symptoms were attributed to other factors [51]. Next, four cases of ATLL were described in individuals initially thought to have non-Hodgkin’s lymphoma. It is therefore during an epidemiological investigation that a new analysis of the medical records of these patients (three women aged 35, 48, and 52, and a 42-year-old man), combined with the new clinical and biological examinations beyond the HTLV-1 seropositivity, had permitted the diagnosis of ATLL. One to three months after being diagnosed, the four patients died, showing the fatality and low survival median that characterize this associated pathology, which is not well-known by local clinicians [52]. Another lymphoma, mycosis fungoides, had also been described in a 58-year-old Gabonese patient, who was equally infected with HTLV-1, and no correlation between the two conditions was established [54]. In addition, the molecular epidemiology of HTLV-1 in Africa also mentioned strains of HTLV-1 from eight Gabonese patients whose clinical status were either ATLL or TSP/HAM [47,48]. The last cases of the HTLV-1-associated diseases in Gabon were reported in 2019, by Igala et al., where three patients (a 50-year-old woman and two men aged 39 and 50-years of age), with ATLL and all were from the southeastern part of the country (Haut-Ogooué province) [63]. Although, the diagnosis of the HTLV-1 infection was retained only in the face to the ELISA positivity, several other clinical and paraclinical examinations performed (lymph nodes, spleen, liver, skin involvement, blood count, immunophenotyping, calcium rate, the value of lactic dehydrogenase, blood smear, and others) have made it possible to retain the diagnosis of ATLL in its acute/chronic and lymphomatous forms for the woman and both men, respectively [63]. However, the death of patients with acute and lymphomatous forms occurred, despite chemotherapy treatments, during hospitalization. Only one patient with the chronic form survived after the treatments. Compared to the chronic form where the overall survival means is about 2 years, the acute forms (leukemia and lymphomatous forms) are more aggressive with a mean lifetime of about 1 to 6 months, regardless of whether chemotherapy treatment is used [61,62]. Thus, since the first study was carried out on the HTLV-1 infection in Gabon, very few diseases associated with this retroviral infection have been reported (Table 2). Several factors could explain the low number of reports of diseases associated with the HTLV-1 infection. Beyond the low life expectancy in Gabon, which is about 62-years, the virus latency can extend over several decades, as well as the low incidence of diseases in infected people, all making the epidemiological monitoring very difficult [61,62]. Most outbreaks of HTLV-1 infections are in isolated zones where people do not necessarily have access to quality medical care, although more than half of the Gabonese population is concentrated in Libreville and its surroundings. Due to those difficulties to access proper healthcare facilities, populations tend to primarily rely on traditional medicine. Disease cases associated with the HTLV-1 infection in Gabon are probably not reported, and therefore highly under-diagnosed, rather than rare or absent. Therefore, the real impact of the HTLV-1 infection on public health in Gabon remains to be fully assessed. Other approaches are needed to properly examine the pathologies associated with HTLV-1 and mostly between individuals from the southeast of the country (Haut-Ogooué), where the most cases have been reported; with 61.54% of pathology cases associated with HTLV-1, to date. The remaining 38.46% of cases (5/13) was diagnosed in Libreville, the capital city, without having a real idea of their provincial origin. Indeed, the case of mycosis fungoid was reported by Perret et al., in a lady from a village in the country, without exactly specifying the place of origin [54]. As well, the four ATLL reported cases by Delaporte et al., in 1993, did not offer an exact idea of the patient’s provincial origin. As the diseases associated with the HTLV-1 infection are very grave pathologies and often uncommon for healthy people, patients with these types of diseases should be transferred to major cities with better healthcare facilities. As mentioned above, the Haut-Ogooué province is the 3rd most endemic province in the country for this retrovirus with a prevalence rate of 9.5 to 11.6%. The high prevalence rate in this province could explain the high number of cases reported in this region of the country, but it should also be important to highlight that the majority (5/8 reported) of cases were described in different epidemiological studies in this region of the country [47,48,51]. Finally, the HTLV-1 infection is associated with several other pathologies, such as systemic lupus, polymyositis, arthrosis, Sjörgen syndrome, uveitis, or vasculitis [64,65,66]. In Gabon, none of these pathologies have been yet reported in individuals infected with HTLV-1. Moreover, in-depth studies on this topic should allow for a better assessment of the impact of this retrovirus on the health of the populations.

### 3.4. Molecular Epidemiology of HTLV-1 in Gabon

The first molecular biology analyses on the strains of HTLV-1 began in 1987. Following the culture of lymphocytes from a healthy HTLV-1 carrier, the provirus was analyzed using a southern blot. The provirus genome of the studied strain showed about 38% of differences between its restriction sites and those of the Japanese prototype. In addition, its restriction map looked similar to that of a STLV-1 (isolated from a Sierra Leonean chimpanzee) and a B subtype HTLV-1 from Zaire [67]. These results were confirmed by sequencing the same strains after the amplification of a region in the LTR gene [70]. However, after sequencing the complete *Env* gene, the new Gabonese strains were classified into a new group (subtype D) of African strains [71]. The sequencing of the *tax*, *env*, and *LTR* genes of the HTLV-1 strains from pregnant women or blood donors found in the HTLV-1 strains of subtype B, but also one belonging to the cosmopolitan subtype A [45,68]. Thus, the Gabonese strains belong to distinct groups, according to the phylogeny analyses: subtype A cosmopolitan, subtype B of Central Africa (consisting of strains from Zaire, Cameroon, and the Central African Republic), and the new subtype D of Central Africa, also consisting of strains of pygmies from Cameroon and the Central African Republic. These new strains (HTLV-1B) were also found to be close to STLV-1 of the great chimpanzee apes [45,48]. This rapprochment reinforces the hypothesis of the inter-species transmission between apes and humans, putting forward several seroprevalence studies [48,69]. In addition, there was the case of a little girl infected with HTLV-1D, whose father was negative for the screening of HTLV-1 and the mother was positive for HTLV-1B [46]. An interesting study on some strains with an indeterminate serological profile has made it possible to classify one of them among HTLV-1 of the molecular subtype F (HTLV-1F) and the others with subtype B [69]. This HTLV-1F formed a clade with baboon monkey STLVs (*Papio* genus).

Finally, another study on the whole sequences of the envelope genes, compared with all of those of PTLV-1 available in the GenBank, showed that the Gabonese strains HTLV-1 belong to subtypes B of Central Africa, cosmopolitan A and with characteristic groups, according to the mutations observed [47]. The phylogenetic analyses showed that the Gabonese strains are divergent from each other although they are in the same geographical region [47,72]. According to the specific mutations of these strains, some subgroups were formed, and the characterization of these subtypes, groups, or strains should not be blindly assumed but by associating the characteristic mutations of a given region. Figure 2 provides an overview of the distribution of HTLV-1 in Gabon, according to prevalence of each province and the described viral subtypes.

## 4. HIV-1 and HTLV-1 Co-Infections in Gabon

Despite the description of both retroviruses, only one study was recently published on the co-infections in populations living in Gabon [37]. A total of 299 individuals (mean age 46 years), including 90 men (30%) and 209 women (70%), were recruited. All were followed up at the Ambulatory Treatment Centre of Franceville, Haut-Ogooué province, and under ART. Of these, 45 were ELISA HTLV-1/2 seropositive and, according to the western blot criteria, 21 out of 45 were confirmed positive; 20 were positive for HTLV-1 (44%), and one for HTLV-1/2 (2%), two were indeterminate (4%), and 22 were seronegative (49%). The PCR results showed that 23 individuals were positive for the tax/rex region. Considering both serological and molecular assays, the prevalence rate of the HTLV-1 infection was estimated at 7.7%, among the HIV-1 patients. Gender and age were identified as independent risk factors. The risk of co-infection was directly proportional to the increasing age of women. The mean CD4+ cell counts were higher in HTLV-1/HIV-1 co-infected (578.1 ± 340.8 cells/mm^3^) than in the HIV-1 mono-infected (481.0 ± 299.0 cells/mm^3^) individuals. Similarly, the mean HIV-1 viral load was Log 3.0 ± 1.6 copies/mL in mono-infected individuals and Log 2.3 ± 0.7 copies/mL in co-infected individuals.

The targets of the WHO, relative to epidemiological surveillance and HIV eradication by 2030 through the 95-95-95 program, could be hampered by some infections, such as HTLV-1, which have the ability to counteract and negatively modulate the survival prognosis of PLHIV [73,74]. HTLV-1, on the one hand, shares the same tropism (CD4+ T) with HIV-1, but, also on another hand, the same transmission routes lead to the frequent observation of HIV-1/HTLV-1 co-infection cases in endemic areas [8,75]. Indeed, studies suggested that the initial infection with HTLV-1 increases the risk of contracting the HIV-1 infection, and vice versa [76]. In addition, several prospective studies conducted in endemic areas for these retroviruses, showed a higher prevalence for theHTLV-1 infection (up to more than 10%) in HIV-1-positive individuals, compared to HIV-1-negative individuals or in the general population [37,77,78]. That alone necessitates the systematic testing for these retroviruses, when one of these retroviruses has been positively diagnosed in an individual in endemic areas. HTLV-1 research in hospitals is not systematic in Gabon [56], and the associated pathologies are still unknown to many local clinicians. So, this strong association between the HTLV-1 infection and HIV-1, coupled with the lack of knowledge or negligence of diseases associated with HTLV-1 by clinicians, could go unnoticed and therefore constitute an obstacle to achieving the goal of diagnosing 95% of HIV-1 positives by 2030.

Although several studies are reporting both a faster progression to the AIDS phase and the often-higher CD4+ count in HIV-1/HTLV-1 co-infected people, compared to HIV-1 mono-infected individuals alone, nothing seems to be solved about these observations [37,79,80]. HTLV-1 could interact with HIV-1 at the molecular and immunological levels, producing immunosuppression and thus a microenvironment conducive to the development of opportunistic pathologies [81,82]. HIV-1, by its regulatory protein p19 (Rev), could mediate the transcription of the HTLV-1 genome by facilitating the binding of the tax protein (encoded by the pX region of the virus, located between the env gene and the 3′ LTR sequence) at its 5′ LTR promoter region [83]. The tax protein is an oncogenic protein transactivator of several cellular genes involved in the cell immortalization, cell replication, and anti-apoptotic signals, but also in the production of cytokines among many other cell mechanisms. However, it has been shown that the Tax alone cannot bind itself in the 5′ LTR promoter region to induce its own synthesis, as well as that of the other HTLV-1 genes [84,85]. Therefore, it interacts with other transcription factors, such as cAMP responses element binding (CREB), transcription activator factors (ATFs), and others to stimulate the viral transcription [84]. Furthermore, the rev protein of HIV-1 would tend to improve these different interactions and thus stimulate the transactivation of the HTLV-1 genes but also those of several other cellular genes, indirectly [83]. Some studies have demonstrated that the tax protein of HTLV-1 could also act as a HIV-1 transcription transactivating protein (tat) by stimulating the positive b-type transcription elongation factor (P-TEFb). Such a stimulation will activate the latent state HIV-1 in the infected cells through the release of cyclin-dependent kinase 9 (CDK9) and cyclin T1. These released proteins could therefore be responsible for an increase in the HIV-1 viral load and the earlier onset of the AIDS phase, followed by the opportunistic infections, which might be delayed in mono-infected patients with HIV-1 [82]. These studies seem to confirm those conducted in Brazil and Guinea-Bissau, reporting a high prevalence of the *Mycobacterium tuberculosis* infection (nearly 2.5 times higher in Pedral-Sampaio’s study) in HIV-1/HTLV-1 co-infected individuals, compared to the prevalence of the same infection in HIV-1 mono-infected individuals [86,87]. Moreover, unlike mono-infected people with HIV-1 or HTLV-1, several studies on the impact of HIV-1/HTLV-1 co-infections indicated an increased production of the IL-2 and INFγ cytokines and the worsening of symptoms (thrombocytopenia, urinary tract infection, opportunistic infections), as well as the faster development and progression of the pathologies associated with HTLV-1, including TSP/HAM and ATLL [88,89,90]. Various ARTs administered to PLHIV have generally been aimed to stop the viral replication, but also to stimulate the immune system, to allow the elimination of the circulating HIV and other infections potentially present in the patient’s body [91]. Despite the fact that HIV-1 and HTLV-1 are viruses of the same family, the different ART regimens currently proposed do not have a real efficacy against HTLV-1 and the related pathologies [92]. Several integrase inhibitors, such as raltegravir, have been shown to inhibit the infectious spread of HTLV-1 in vitro, thus reducing the possibility of the HTLV-1 transmission from one individual to another [93]. However, in an established infection, HTLV-1 predominantly persists as a provirus integrated into the cellular DNA, so ART has not been shown to reduce the proviral reservoir and the possibility of developing the HTLV-1 associated diseases. Contrary to HIV-1, HTLV-1 has a limited use for its reverse transcriptase (most during the 1st step of infection) so it further duplicates its genetic material by the clonal expansion (mitotic replication of the infected cells) [61,94]. As the role of the integrase inhibitors is the inhibition of the HTLV-1 provirus DNA insertion in the chromatids of hosts cells [93], the integrase inhibitors could not be effective on the HTLV-1 proliferation and its consequences following this step of infection. Furthermore, starting from the fact that the HTLV-1 infection alone can cause immunosuppression, it would therefore be possible that HTLV-1, in HIV-1/HTLV-1 co-infected people under ART, can reduce the effectiveness of the immune system [81], thus preventing the elimination of the circulating HIV-1.

A study by Beilke et al. investigating the relationship between the HTLV-1/2 viral load, clinical parameters in HIV-1 positive patients on ART and its co-infection with HTLV-1 or HTLV-2, found a significant association between the HTLV-1 burden proviral DNA > 20,000/10^6^ peripheral blood mononuclear cells (PBMCs) and a HIV-1 viral burden load < 10,000 copies/mL [95]. This study also reported a much higher CD4+ count and HTLV-1 proviral load in HIV-1/HTLV-1 co-infections than in HIV-1/HTLV-2 co-infections. Although this study was conducted on a relatively small number of individuals (72), it still provides insight into the reality of HIV-1/HTLV-1 co-infections, due to the fact that even on ART, individuals co-infected with HIV-1/HTLV-1 are still at risk of immunosuppression and opportunistic infections, due to the highHTLV-1 turn-load and increased production of different cytokines, contributing to the occurrence of the pathologies associated with HTLV-1 and the worsening of symptoms.

## 5. Conclusions

Several studies have shown a high prevalence of HTLV-1 among HIV-1 infected people, compared to the general population in endemic areas, for both retroviruses. HTLV-1-induced pathologies remain unknown to most medical practitioners. Thus, people living with a disease associated with HTLV-1, could potentially be positive for HIV-1 but could go unnoticed, constituting the first obstacle in the aim of the WHO for 2030. However, many others studies demonstrated the HTLV-1 ability to activate the HIV-1 viral latency, while others reported a high level of proinflammatory cytokines production and a high HTLV-1 viral load in HIV-1/HTLV-1 co-infections, compared to HIV-1/HTLV-2 or HIV-1 and HTLV-1/2 mono-infections. These results highlight the involvement of these co-infections in the worsening of symptoms and the immunosuppression, even on ART, the faster development of the HTLV-1 diseases, and the progression to the AIDS state. Furthermore, in Gabon, as for other endemic areas for both retroviruses, a systematic diagnosis of HTLV-1 should be imposed on HIV-1 positive patients (and vice versa) to evaluate their survival prognosis. In addition, to achieve the 2030 WHO goals, in the endemic area, a systematic screening of HTLV-1 should be imposed, as well, on pregnant women, and blood transfusion should be avoided or limited, as it increases the risk of HIV-1/HTLV-1co-infections. Finally, the use of condoms during sexual intercourse, and screening organs before transplantations must be also implemented to strengthen the fighting strategy against both viruses. These would help in achieving the WHO goals, as the main ways proposed for the HIV-1 elimination by 2030 could be used for the HTLV-1 elimination as well. considering the high contact frequency between humans and NHPs (causes of bites), and the NHP consumption in Gabon, a sensibilization campaign about the risks involved could appear as an important effort to achieve the aims of the WHO, by 2030.

## Figures and Tables

**Figure 1 viruses-14-02808-f001:**
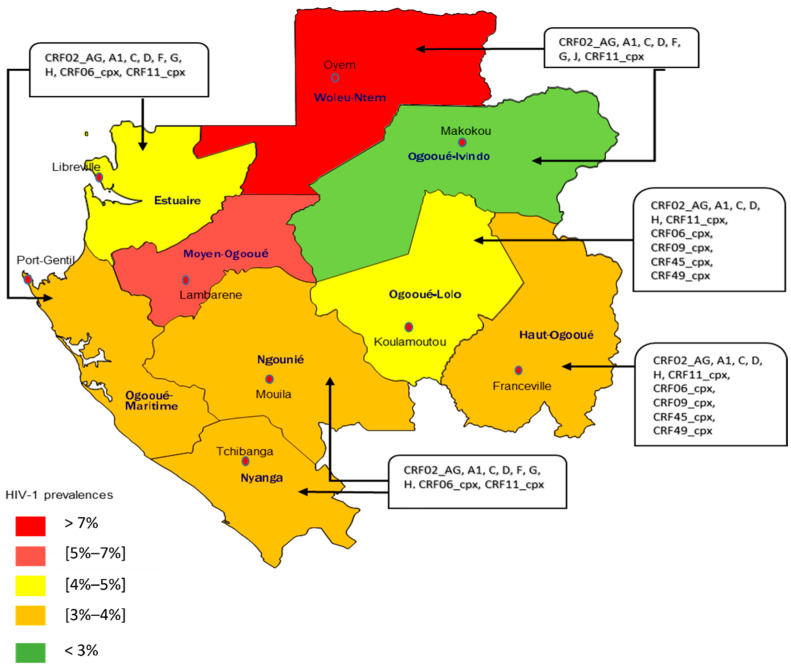
Distribution of HIV-1 subtypes in Gabon and the prevalence by province.

**Figure 2 viruses-14-02808-f002:**
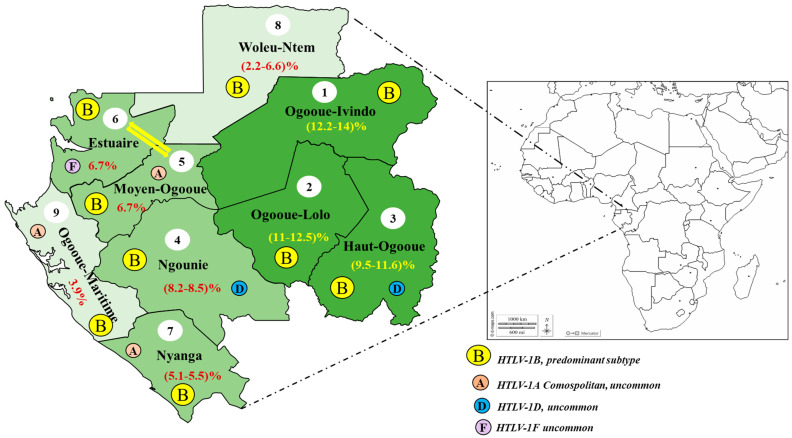
HTLV-1 distribution, according to prevalence of each province and the viral subtype described [11,12,45,46,47,48]. Numbers in brackets are the HTLV-1 prevalence in each province. Thus, the provinces are classified accordingly to the prevalence; number 1 is the highest and 9 is the lowest. Letters indicate the described HTLV-1 subtype in each province.

**Table 1 viruses-14-02808-t001:** Genetic diversity of HIV around Gabon.

Country	Groups and Subtypes	Recombinant Forms	Origin	Reference
Cameroon	HIV-1 Group M, N, O, P A–D, F–H, J, K	CRF01_AE, CRF02_AG, CRF06_cpx, CRF09_cpx, CRF11_cpx, CRF13_cpx, CRF18_cpx, CRF22_01A1cpx, CRF25_cpx, CRF36_cpx, CRF37_cpx	Human immunodeficiency virus type 1 (HIV-1) subtypes in the northwest region, Cameroon	[26]
			HIV infections in Northwestern Cameroon: identification of HIV-1 group O and dual HIV- 1 group M and group O infections	[27]
			Four New HIV-1 group N isolates from Cameroon: Prevalence continues to be low	[28]
			Confirmation of the putative HIV-1 group P in Cameroon	[29]
	HIV-2		HIV-2 intergroup recombinant identified in Cameroon	[30]
Equatorial Guinea	Group M A, B, C, D, E, F, G, H	CRF02_AG, CRF06_cpx, CRF09_cpx, CRF11_cpx CRF18_cpx	Description of the HIV-1 group M molecular epidemiology and drug resistance prevalence in Equatorial Guinea from migrants in Spain	[31]
Democratic Republic of Congo	Group M A1, A2, B, C, D, F1, G, F, J, K, L	CRF02_AG, CRF13_cpx, CRF25_cpx, CRF26_cpx,	HIV-1 subtypes and drug resistance mutations among female sex workers varied in different cities and regions of the Democratic Republic of Congo	[32]

**Table 2 viruses-14-02808-t002:** Cases of pathologies associated with the HTLV-1 infection in Gabon.

Pathologies	Area of Diagnosis	Provincial Origin of the Patients	Sex	Age (by Year)	Outcome Diagnosis	Year of Diagnosis	References
TSP/HAM	Franceville	HAUT-OGOOUE	M	55	ND	1989	[67]
TSP/HAM	Franceville	HAUT-OGOOUE	M	60	ND	1989	[67]
TSP/HAM?	Ayanabo	HAUT-OGOOUE	M	≈48	ND	2005	[68]
TSP/HAM	Franceville	HAUT-OGOOUE	F	70	ND	1997	[69]
TSP/HAM	Franceville	HAUT-OGOOUE	F	…	ND	1997	[69]
ATL	Libreville	ESTUAIRE?	F	35	LNH	1989	[70]
ATL	Libreville	ESTUAIRE?	F	48	LNH	1989	[70]
ATL	Libreville	ESTUAIRE?	F	52	LNH	1989	[70]
ATL	Libreville	ESTUAIRE?	M	42	LNH	1989	[70]
ATL?	Libreville	…	M	58	MF	1996	[71]
ATL	Libreville	HAUT-OGOOUE	F	50	LNH	2019	[72]
ATL	Libreville	HAUT-OGOOUE	M	50	LNH	2019	[72]
ATL	Libreville	HAUT-OGOOUE	M	39	LNH	2019	[72]

ATL: adult T cell leukemia/lymphoma; TSP/HAM: tropical spastic paraparesis/HTLV-1 associated myelopathy; ND: undiagnosed; LNH: Non-Hodgkin’s lymphoma; MF: mycosis, fungoides; …: not specified; ?: not sure.

## Data Availability

Not applicable.

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
