# Peer review of "Human T-Cell Leukemia Virus Type 1-Related Diseases May Constitute a Threat to the Elimination of Human Immunodeficiency Virus, by 2030, in Gabon, Central Africa"

_viruses, 2022, doi:10.3390/v14122808_

Round 1
Reviewer 1 Report
Review of ‘Human T-Lymphotropic Virus Type 1 -Related Diseases May Constitute a Threat to the Elimination of Human Immunodeficiency Virus by 2030 in Gabon, Central Africa’.
This manuscript reviews what is currently known about the epidemiology of HTLV-1 in Gabon, and asks whether coinfection with HIV/HTLV-1 might increase the probability of developing diseases related to both viruses.
This paper is timely: last year, the world health organisation has highlighted HTLV-1 as a strategic priority area for public health intervention. Epidemiological data indicates that HTLV-1 prevalence is high in Gabon, thus a high prevalence of HTLV-1 associated disease is also to be predicted. However the reported rates of HTLV-1 –related diseases are low, which is most likely explained by underdiagnosis of HTLV-1-associated disease and a lack of awareness of the virus. Therefore, there is a strong justification to publish this paper to promote awareness and future work on this previously neglected public health need.
I have some minor concerns to raise with the authors.
1. Minor but important clarifications are needed to the section on antiretrovirals (line 467-471. Several antiretroviral therapies have been shown to block infectious spread of HTLV-1 in vitro (e.g. Raltegravir Barski et al, frontiers in microbiology 2019). So, antiretrovirals may be very effective at blocking mother to child transmission of HTLV-1. However, in established infection, as HTLV predominantly persists as a provirus integrated into cellular DNA, antiretroviral therapy has not been shown to reduce the proviral reservoir or reduce the chance of developing HTLV-1 associated diseases.
2. Given the theme of the paper, it would be useful to discuss the strategies for elimination of HTLV-1 in the conclusion section, e.g. by preventing mother to child transmission, use of condoms, screening blood for transfusion and organs for transplantation. Might HTLV-1 also be eliminated using a similar strategy to that proposed for HIV?
Minor concerns
1. Abstract Human T cell Leukaemia virus type-1 or Human T cell lymphotropic virus type 1 is the correct name for HTLV-1
2. Discussing the expected number of cases of HTLV-1 related diseases (line328) should incorporate time- (e.g. cases per year).
3. HTLV-2 is briefly mentioned however it would be worth a sentence or two on the topic to explain why HTLV-2 is not a concern in this context.
4. Line 229 concerns the timing at which HTLV-1 infects babies. The data presented does not conclusively prove the timing at which babies become infected, so it is better to soften the statement.
Grammatical errors
1. Figure 2 legend HTLV1F is mislabelled.
2. Line 474 should read mononuclear
Author Response
Reviewer #1:
Comments and Suggestions for Authors
Review of ‘Human T-Lymphotropic Virus Type 1 -Related Diseases May Constitute a Threat to the Elimination of Human Immunodeficiency Virus by 2030 in Gabon, Central Africa’.
This manuscript reviews what is currently known about the epidemiology of HTLV-1 in Gabon, and asks whether coinfection with HIV/HTLV-1 might increase the probability of developing diseases related to both viruses.This paper is timely: last year, the world health organisation has highlighted HTLV-1 as a strategic priority area for public health intervention. Epidemiological data indicates that HTLV-1 prevalence is high in Gabon, thus a high prevalence of HTLV-1 associated disease is also to be predicted. However the reported rates of HTLV-1 –related diseases are low, which is most likely explained by underdiagnosis of HTLV-1-associated disease and a lack of awareness of the virus. Therefore, there is a strong justification to publish this paper to promote awareness and future work on this previously neglected public health need.
I have some minor concerns to raise with the authors.
- Minor but important clarifications are needed to the section on antiretrovirals (line 467-471. Several antiretroviral therapies have been shown to block infectious spread of HTLV-1 in vitro (e.g. Raltegravir Barski et al, frontiers in microbiology 2019). So, antiretrovirals may be very effective at blocking mother to child transmission of HTLV-1. However, in established infection, as HTLV predominantly persists as a provirus integrated into cellular DNA, antiretroviral therapy has not been shown to reduce the proviral reservoir or reduce the chance of developing HTLV-1 associated diseases.
The sentences have been revised and changed accordingly. Clarifying statements have been added about raison why inhibitor/raltegravir is just as prophylactic treatment (from line 506 to 514).
- Given the theme of the paper, it would be useful to discuss the strategies for elimination of HTLV-1 in the conclusion section, e.g. by preventing mother to child transmission, use of condoms, screening blood for transfusion and organs for transplantation. Might HTLV-1 also be eliminated using a similar strategy to that proposed for HIV?
The suggestion has been considered and added (line: 542 to 547). About the question whether HTLV-1 might also be eliminated using a similar strategy to that proposed for HIV, it is yes, most probably because both viruses share same ways of transmission; a systematic screening of pregnancy women, donors, the use of condoms could be helpful to reduce the risk of infection of both viruses.
Minor concerns
- Abstract Human T cell Leukaemia virus type-1 or Human T cell lymphotropic virus type 1 is the correct name for HTLV-1
For this suggestion, Human T cell leukemia virus type 1 was adopted (line 18).
- Discussing the expected number of cases of HTLV-1 related diseases (line328) should incorporate time- (e.g. cases per year).
In this review, the number was estimated from an estimate of infected people in Gabon (made by Gessain and Cassar in 2012) and the risk of developing an associated disease to HTLV-1 (3-7%). Clarifications were added accordingly (lines 365 to 369).
- HTLV-2 is briefly mentioned however it would be worth a sentence or two on the topic to explain why HTLV-2 is not a concern in this context.
The suggestion had been considered and statement added at the end of introduction (line 58 to 62)
- Line 229 concerns the timing at which HTLV-1 infects babies. The data presented does not conclusively prove the timing at which babies become infected, so it is better to soften the statement.
Additional details have been given to clarify the statement, which was revised to make it soften and the seroconversion seem to happen after 18 months of age (line 293-313).
Grammatical errors
English editing has been done thoroughly.
- Figure 2 legend HTLV1F is mislabelled.
This suggestion has also been taken into account. The letter D has replaced by F and indicated accordingly in the legend.
- Line 474 should read mononuclear
We changed “monocular” by “mononuclear” (line 520)
Reviewer 2 Report
The authors sought to summarize the previous findings regarding HIV-1 and HTLV-1 infection and their clinical implication in Gabon, followed by the significance of their coinfection. In general, this manuscript clearly and concisely reviewed and I think that it is well comprehensive and should be helpful for the many clinical doctors who take care of HIV-1 and HTLV-1 carriers in Africa as well as other countries. Most of my comments as shown below are minor but the authors must carefully check thoroughly the grammatical errors and typos, as well as the format in the References.
Regarding HTLV-1, the authors described in a lot of different ways such as human T-lymphotropic virus type-1 (line 2), human T-cell lymphoma virus type 1 (line 18), human T-lymphotropic virus (line 164), or human T-cell leukemia virus of type 1 (line 173). These must be unified carefully; I believe that human T-cell leukemia virus type 1 is general.
Line 25; I feel ‘Moreover’ appears to be unnecessary.
Line 45; HTLV-1 should be fully spelled out here (this is the first time to use in the text) but not later.
Line 49; Tropical Spastic Paraparesis/HTLV-1 Associated Myelopathy-underline parts should not be capital letters
Line 60; Abbreviation of ‘Human immunodeficiency virus’ was described above in line 32.
Line 72; ‘These findings suggested an evolution of HIV prevalence in Gabon since 2012.’, what do you mean by this?
Line 107 and thereafter; I would request to describe the prevalence of HIV-2 in comparison with HIV-1.
Table 1; Are VIH-1 and VIH-2 right or HIV-1 and HIV-2? This table has many other typos such as Human immunodeficiency virus type 1 ((HIV-1), coexistence of HIV type 1 and HIV-1, etc.
Line 272; The authors mentioned that no child born from seropositive mothers presented a seroconversion after one year, while they also mentioned that this finding was consistent with other studies from the same team which described a 2.8% rate of HTLV-1 seroprevalence in children. I think these descriptions include clear discrepancy. The authors should reconsider this discrepancy. Entirely, I am afraid that the contents of this subsection were not clearly described and sometimes inconsistency was seen.
Figure 2; brakets should be brackets. The percentages were shown either with or without bracket. 6,7% should be 6.7%.
Line 325 and Table 2; already should be previously or eliminated.
Table 2; The abbreviation of HAM should be amended. 019 in the last line of Table should be 2019.
Line 397; The relation ship of zoonotic transmission from chimpanzees in the cases of other regions than Gabon should be described to help understand the entire HTLV-1 zoonotic transmission from non-human primates.
Author Response
Reviewer #2:
The authors sought to summarize the previous findings regarding HIV-1 and HTLV-1 infection and their clinical implication in Gabon, followed by the significance of their coinfection. In general, this manuscript clearly and concisely reviewed and I think that it is well comprehensive and should be helpful for the many clinical doctors who take care of HIV-1 and HTLV-1 carriers in Africa as well as other countries. Most of my comments as shown below are minor but the authors must carefully check thoroughly the grammatical errors and typos, as well as the format in the References.
English editing had been done thoroughly and references double-checked.
Regarding HTLV-1, the authors described in a lot of different ways such as human T-lymphotropic virus type-1 (line 2), human T-cell lymphoma virus type 1 (line 18), human T-lymphotropic virus (line 164), or human T-cell leukemia virus of type 1 (line 173). These must be unified carefully; I believe that human T-cell leukemia virus type 1 is general.
For this suggestion, Human T cell leukemia virus type 1 was adopted (line 18).
Line 25; I feel ‘Moreover’ appears to be unnecessary.
For this suggestion, ‘moreover’ was delete (line 25)
Line 45; HTLV-1 should be fully spelled out here (this is the first time to use in the text) but not later.
As suggested, HTLV-1 by Human T cell Leukemia Virus type 1 was fully spelt (line 45).
Line 49; Tropical Spastic Paraparesis/HTLV-1 Associated Myelopathy-underline parts should not be capital letters
Capital letters have been removed. So, we written ‘tropical spastic paraparesis/HTLV-1 associated myelopathy. We’ve also removed capital letters for adult T cell leukemia virus. (Line 49-50)
Line 60; Abbreviation of ‘Human immunodeficiency virus’ was described above in line 32.
Here we changed Human immunodeficiency virus by HIV-1 (line 65)
Line 72; ‘These findings suggested an evolution of HIV prevalence in Gabon since 2012’, what do you mean by this?
We meant an increase. Thus, the statement was revised, and modification was made with additional description (line 76-78)
Line 107 and thereafter; I would request to describe the prevalence of HIV-2 in comparison with HIV-1.
The statement has been considered, but the statement has been clarified (line 113-116). In Gabon, since the study by Delaporte et al in 1996 which reported a low seroprevalence of HIV-2 (3.5% or 8/226 samples), no other study has focused on the prevalence of HIV-2. Also, knowing that Gabon is located in Central Africa where the M group of HIV-1 predominates (Giovanetti et al., 2020; Hemelaar, 2012; Hemelaar et al., 2020), all the studies carried out in this country have focused exclusively on HIV-1 infection.
Table 1; Are VIH-1 and VIH-2 right or HIV-1 and HIV-2? This table has many other typos such as Human immunodeficiency virus type 1 ((HIV-1), coexistence of HIV type 1 and HIV-1, etc.
All types have been changed in table 1 as VIH-1 and VIH-2 have been changed to HIV-1 and HIV-2 respectively. We also deleted one bracket for (HIV-1) and HIV.
Line 272; The authors mentioned that no child born from seropositive mothers presented a seroconversion after one year, while they also mentioned that this finding was consistent with other studies from the same team which described a 2.8% rate of HTLV-1 seroprevalence in children. I think these descriptions include clear discrepancy. The authors should reconsider this discrepancy. Entirely, I am afraid that the contents of this subsection were not clearly described and sometimes inconsistency was seen.
Additional details have been given to clarify the statement, which was revised to make it clearer, and the seroconversion seems to happen after 18 months of age (line 293-313).
Figure 2; brakets should be brackets. The percentages were shown either with or without bracket. 6,7% should be 6.7%.
Thank you so much for these suggestions. Here, the correction was made and for prevalence, brackets were included, and commas replaced by full stops.
Line 325 and Table 2; already should be previously or eliminated.
For title of this table, “already” was eliminated.
Table 2; The abbreviation of HAM should be amended. 019 in the last line of Table should be 2019.
As suggested, abbreviations of HAM has been replaced by HTLV-1 associated myelopathy and the number corrected.
Line 397; The relationship of zoonotic transmission from chimpanzees in the cases of other regions than Gabon should be described to help understand the entire HTLV-1 zoonotic transmission from non-human primates.
The suggestion has been considered and additional information added to better the understanding (line 345-359)